# Rheumatic Heart Disease Screening Based on Phonocardiogram

**DOI:** 10.3390/s21196558

**Published:** 2021-09-30

**Authors:** Melkamu Hunegnaw Asmare, Benjamin Filtjens, Frehiwot Woldehanna, Luc Janssens, Bart Vanrumste

**Affiliations:** 1eMedia Research Lab/STADIUS, Department of Electrical Engineering (ESAT), KU Leuven, Andreas Vesaliusstraat 13, 3000 Leuven, Belgium; benjamin.filtjens@kuleuven.be (B.F.); luc.janssens@kuleuven.be (L.J.); bart.vanrumste@kuleuven.be (B.V.); 2Center of Biomedical Engineering, Addis Ababa Institute of Technology, Addis Ababa University, Addis Ababa P.O. Box 385, Ethiopia; f.woldehanna@aau.edu.et; 3Intelligent Mobile Platforms Research Group, Department of Mechanical Engineering, KU Leuven, Andreas Vesaliusstraat 13, 3000 Leuven, Belgium

**Keywords:** phonocardiogram, nested cross-validation, acoustic features, rheumatic heart disease, screening, support vector machines

## Abstract

Rheumatic heart disease (RHD) is one of the most common causes of cardiovascular complications in developing countries. It is a heart valve disease that typically affects children. Impaired heart valves stop functioning properly, resulting in a turbulent blood flow within the heart known as a murmur. This murmur can be detected by cardiac auscultation. However, the specificity and sensitivity of manual auscultation were reported to be low. The other alternative is echocardiography, which is costly and requires a highly qualified physician. Given the disease’s current high prevalence rate (the latest reported rate in the study area (Ethiopia) was 5.65%), there is a pressing need for early detection of the disease through mass screening programs. This paper proposes an automated RHD screening approach using machine learning that can be used by non-medically trained persons outside of a clinical setting. Heart sound data was collected from 124 persons with RHD (PwRHD) and 46 healthy controls (HC) in Ethiopia with an additional 81 HC records from an open-access dataset. Thirty-one distinct features were extracted to correctly represent RHD. A support vector machine (SVM) classifier was evaluated using two nested cross-validation approaches to quantitatively assess the generalization of the system to previously unseen subjects. For regular nested 10-fold cross-validation, an f1-score of 96.0 ± 0.9%, recall 95.8 ± 1.5%, precision 96.2 ± 0.6% and a specificity of 96.0 ± 0.6% were achieved. In the imbalanced nested cross-validation at a prevalence rate of 5%, it achieved an f1-score of 72.2 ± 0.8%, recall 92.3 ± 0.4%, precision 59.2 ± 3.6%, and a specificity of 94.8 ± 0.6%. In screening tasks where the prevalence of the disease is small, recall is more important than precision. The findings are encouraging, and the proposed screening tool can be inexpensive, easy to deploy, and has an excellent detection rate. As a result, it has the potential for mass screening and early detection of RHD in developing countries.

## 1. Introduction

Rheumatic heart disease (RHD) remains a number one cause of preventable cardiovascular diseases (CVDS) [1]. It is also the major cause of heart disease for children and teenagers in low and middle-income countries and several marginalized groups in developed countries, causing a major health challenge for them [2]. RHD is a manifestation of poverty, overcrowded housing, and poor hygiene [3]. Worldwide, over three-quarters of children aged under 15 years old live in the endemic regions where RHD causes a serious health problem [4].

In terms of its pathologic pathway, RHD is a sequel to acute rheumatic fever (ARF). ARF itself is an autoimmune response for a group A streptococcal (GAS) bacterial infection. The repeated GAS infections lead to considerable valve damage, which will lead to RHD [5]. GAS mainly affects school-going children. The popular belief in the past was that there is a considerable latency time between the first ARF and RHD. However, current studies have demonstrated that RHD manifests itself as early as one year from the first episode of ARF [6]. The shorter latency was more prevalent in African countries where the disease burden is high with significant mortality and morbidity at adolescent age [7].

RHD often presents as moderate to severe valve damage, heart failure, pulmonary hypertension, atrial fibrillation, stroke, and ineffective endocarditis [8]. In the regions where RHD is endemic, late interventions, which include open-heart surgery and valve replacement, are not readily available due to the cost and unavailability of heart surgeons. Primordial prevention is perhaps the best solution. However, effective implementation in developing countries is hampered by significant barriers. The major barriers include low health literacy, higher poverty rates, harmful health-related cultural and social norms, and disparities in language and education. Primordial preventative measures against GAS infection are also inextricably linked to the developmental stages of the countries involved. Secondary prophylaxis, which requires long-term benzathine penicillin G(BPG) injections, is also not reliable due to the hesitation and low adherence of the subjects. However, currently, it is the only working therapeutic solution in RHD endemic nations [9].

When the heart valves are affected by RHD, they will start to leak as they lose their mechanical flexibility to properly open and close. As the disease progresses, it causes the valves to be stiff, eroded, fibrocystic, and the tissues to become loose and flappy [5]. This creates a turbulent blood flow inside the heart chambers which is called a heart murmur. This can be detected by analyzing the heart sound from a phonocardiogram (PCG). The heart sound normally has four fundamental components: the first heart sound (S1), systole, the second heart sound (S2), and diastole. At normal conditions, the systole and diastole intervals are silent, but when RHD occurs, there will be a murmur that can be detected on a phonocardiogram. For optimal listening and recording, the phonocardiogram must be recorded while there is complete silence in the room. Figure 1 shows a graphical representation of heart sounds from a PCG of healthy control (HC) and a person with RHD (PwRHD).

Instead of manual auscultation, the World Heart Federation (WHF) has adopted echocardiographic imaging as a standard diagnostic mechanism for RHD and produced a guideline [10]. This guideline classified an echocardiogram from RHD endemic regions as normal (no RHD), definite RHD, or borderline RHD. Mild pathological aortic or mitral regurgitation, or morphological alterations of the mitral valve alone, are instances of borderline RHD. Borderline cases are usually difficult to detect using manual auscultation. Records show that there is a hesitancy and debate by physicians regarding the requirement of the follow-up and treatment (secondary prophylaxis) of borderline RHD cases [11].

Over the years, the lack of a reliable estimate of the burden of RHD on a global, national, and local scale has been a major obstacle to the implementation of evidence-based, cost-effective RHD prevention methods [12]. J.J. Noubiap et al. [13] undertook a systematic literature review of publications from 1996 to 2017 to establish a current estimate of the global burden of RHD from RHD prevalence studies which were conducted by echocardiographic screening and were population based. For those that employed WHF criteria, World Health Organization (WHO) criteria, and other criteria, the average RHD prevalence was estimated to be 26.1 cases per 1000, 11.3 cases per 1000, and 5.2 cases per 1000, respectively. According to these findings, they indicated that RHD prevalence has remained high. Its prevalence was shown to be highest in Sub-Saharan countries.

For instance, several studies have found a consistent trend of high prevalence rates of RHD in different parts of Ethiopia [14,15,16,17,18]. The recent reported RHD prevalence in Ethiopia was reported by Gemechu T. et al. [19]. They conducted a population-based study based on echocardiographic screening using WHF criteria on participants aged 6 to 25 years. They reported a prevalence of 37.5 cases per 1000 for definite RHD. The prevalence increased to 56.7 cases per 1000 when borderline cases were included. The highest prevalence of 60 cases per 1000 for definite RHD was reported in the 16 to 20 age group. This indicates that the prevalence rate of RHD in Ethiopia is significantly higher than the world average (26.1 cases per 1000).

Developing countries have been focusing on therapeutic interventions in light of these high prevalence rates. However, secondary prevention strategies heavily rely on appropriate case detection. Recently, the advent of echocardiography has opened a new era of transthoracic imaging for successful visualization and quantification of valve damage [20]. However, its cost and requirement of highly trained physicians hindered its application for community-level case detection and screening applications. Therefore, an alternative, cheaper, and robust strategy in the detection of RHD is highly sought.

Godown J. et al. [21] had compared the performance of low-cost handheld echocardiography and auscultation in RHD detection in Uganda. The process involved experienced cardiologists performing auscultation and reading the echocardiography images. A group of 1317 subjects (54% female) underwent echocardiographic screening using a low-cost device and auscultation. There was a 3.4% prevalence rate for RHD and a 9.6% prevalence rate when borderline RHD cases were included. Standard portable echocardiography using the WHF criteria was used to create the ground truth. The auscultation process achieved a sensitivity of 22.2% and specificity of 91.2%, while the low-cost handheld echocardiography achieved a sensitivity of 97.8% and specificity of 87.3% for definite RHD. Thus, it can be observed that the manual auscultation performance was significantly low for RHD detection. However, both techniques still required the involvement of expert cardiologists for screening tasks, which makes the process less favorable in developing countries.

More recently, Gardezi S.K.M. et al. [22] also quantified a result with a similar trend. They reported a sensitivity of 32% and specificity of 67% when auscultation was used to detect mild murmur. For significant murmur, the sensitivity increased to 44%, and the specificity was reported at 69%. In this study, auscultation was also performed by general practitioners and cardiologists.

Ploutz M. et al. [23] considered the involvement of non-expert nurses in echocardiography screening tasks. The nurses were trained on handheld echocardiography for RHD screening and were asked to screen 956 students for RHD. In the study, 913 of the participants were HC, and 43 were PwRHD (32 borderline and 11 definite RHD). Their findings were compared with findings from experts. A sensitivity of 74.4% and a specificity of 78.8% for both definite and borderline RHD were reported. Sensitivity improved to 90.9% for definite RHD. They concluded that the reported results are a reasonable performance involving nurses in the screening programs.

The process of manual cardiac auscultation has been reported to have low sensitivity and specificity in the detection of RHD [21,24]. This is mainly because there is a limited auditory sensitivity for heart sounds due to insufficient training, the presence of noise, and inaccurate conduction [25]. Complementing these limitations with a computer-assisted system from an electronic heart sound record would have a significant advantage for mass screening applications. In the past, multiple studies have been conducted in automatic heart sound analysis. Montinari M. R. et al. [26] have investigated the progress of auscultation in the last 200 years. They reported that the skill for manual auscultation has diminished over the years and concluded that there should be a more balanced approach that integrates the exiting clinical method with a digital system to improve accuracy.

The pioneer reported work that employed computers for cardiac murmur detection was conducted by Gerbarg et al. [27] in the early 1960s. They used threshold-based techniques to locate potential segments with a murmur that are caused by mitral valve insufficiency due to RHD. They have not found any conclusive results but indicated the potential of the computing system for heart sound analysis. Computer-based heart sound analysis has garnered substantial interest in recent years as a result of the introduction of electronic stethoscopes with recording capability.

Automatic heart sound classification algorithms usually localize the fundamental heart sounds (FHS) (S1, S2, systole, and diastole) before classification [28,29,30,31]. In most cases, the incorrect localization of the FHS was partially blamed for the poor classification performance. Based on this, researchers proposed a classification approach without the requirement of segmentation [32,33,34,35,36]. This paper proposes a classification system for heart sounds that does not require segmentation.

Feature extraction for the heart sound classification task was another domain that was heavily explored. The frequently used features for heart sound classification were time domain features [37,38]; frequency domain features [39,40,41]; spectrogram features [42,43,44,45]; a combination of time, frequency and perceptual features [46,47,48,49,50,51,52,53]. A combination of perceptual and wavelet features has also been used [28,54]. There is a significant overlap in the set of features used in the literature.

Careena P. et al. [38] extracted variance, standard deviation, entropy, peak amplitude, RMS, crest factor, impulse factor, shape factor, energy, and clearance factor as time–domain features to classify heart sounds. They argued that time–domain features are analytically straightforward to compute. Their algorithm was evaluated on a public dataset and resulted in good classification performance. A. M. Alqudah et al. [41] used instantaneous frequency features, where the instantaneous frequency of a nonstationary signal was defined as a time-varying parameter that is related to the mean of the frequencies in the signal as it develops. It was computed as the first conditional spectral moment of the power spectrum. They extracted eight different frequency domain features to capture the instantaneous frequency properties of heart sound signals.

M. Deng et al. [45] capitalized on the Mel Frequency Cepstral Coefficients (MFCC), which features excellent performance in many sound perceptions, and they tried to improve the MFCC features to elaborate the dynamic characteristics among consecutive heart sound signals. They argued that, due to the higher sensitivity of the human ear to the dynamic characteristics of acoustic signals, the dynamic information contained in the heart sounds spectrum also offers a wealth of information that can be used to further improve classification accuracy. The first and second derivatives of the MFCC coefficients were also included as features to reflect the dynamic information in the heart sound signal. N. K Sawant et al. [53] proposed time domain features that include zero-crossing rate, the energy of the raw signal and entropy; frequency domain features that include spectral centroid, spectral spread, spectral flux, and spectral entropy; and perceptual features that incorporate the first seven MFCC coefficients.

Springer et al. [28] proposed an automated detection of a specific heart murmur caused by RHD. They developed an SVM-based classification algorithm, which was tested on an in-house dataset and collected from PwRHD and control groups in South Africa. They used undecimated wavelet transform (UWT) and MFCCs in their feature extraction stage. They claimed that UWT was selected over the other wavelet transforms as it provides a wavelet decomposition at discrete decomposition levels, leading to a reduction in the feature space, while not halving the number of samples in the signal at each decomposition level. They used the level 4 Daubechies 5 (db5) wavelet and extracted detail coefficients from level 2, level 3, and level 4. To reduce the feature space, they used the median of a specified window. Their experiment enrolled 109 people. Sixty-nine were HC and the remaining 40 were PwRHD (15 were definite RHD and 25 were borderline RHD). They used an SVM classifier using wavelet and MFCC features where the classification performance was evaluated across 10 iterations of 10-fold cross-validation. They reported accuracy of 74.5 ± 2.1% and an f1-score of 69.9 ± 2.8%. They claimed that the reported low performance was caused by the low-quality dataset they utilized.

After that, Asmare M. H. et al. [44,52] also proposed heart sound classification to detect murmur from RHD. They extracted 26 features from the entire heart sound signal to properly deal with systolic as well as diastolic murmurs caused by RHD [52]. Their features consisted of the time domain, frequency domain, and perceptual domain features. They reported a classification accuracy of 97.1%, a sensitivity of 98%, specificity of 95.3%, and precision of 97.6%. However, their results were based on data collected from a cardiology clinic, where there were more pathological cases than normal, which do not reflect the actual prevalence rate of RHD in the community.

This paper proposes a new and robust RHD screening tool from phonocardiogram recordings. The original contributions of this manuscript are the development of an extensive RHD heart sound dataset, development of a screening tool that can be implemented in an imbalanced dataset with better RHD detection capability, detection of murmur without the need for heart sound segmentation, and a proposal of a unique set of features which appropriately model heart murmur.

The rest of the manuscript is structured as follows. The materials and methods are addressed in Section 2. Section 3 presents the results. Section 4 provides a discussion of the findings, and finally, Section 5 presents the concluding remarks.

## 2. Materials and Methods

The overall methodology of this paper contains four principal components as shown in Figure 2: (1) An extensive dataset was created by combining in-house collected data with data from a freely available public database. (2) Each record was bandpass filtered and normalized. A single 30-s record was extracted from each subject and each record was downsampled to 2 kHz. (3) Thirty-one features were extracted from the preprocessed record. (4) The features were used to train an SVM classifier and data partitioning was performed along with hyperparameter optimization. A detailed description of each block is discussed in the following sub-sections.

### 2.1. Heart Sound Dataset

There was no publicly available heart sound dataset specifically for RHD. Hence, we collected new and extensive heart sound data from PwRHD and HCs as explained below.

#### 2.1.1. RHD Dataset

This data was collected at Tikur Anbessa Referral Teaching Hospital (TASH), one of the biggest hospitals in Ethiopia. The duration of the data collection was from August 2018 to July 2019. Thinklabs One™ electronic stethoscope [55] was used for the data collection (Figure 3a). The stethoscope had a flat response from 20 Hz to 2 kHz. The sampling rate of the device was 44.1 kHz. The stethoscope can be directly connected to a computer or a mobile phone for recording, and it had an earphone integration to listen to the heart sound directly.

Before data collection, each volunteering subject was checked by a cardiologist using transthoracic echocardiography to confirm the presence or absence of RHD (Figure 3b). The heart sound data was then recorded in a real-world clinical environment (Figure 3c). The real-world environment had people talking, instrument noise, movement artifacts, mobile phone interference, and other commonly observed physiological noises. Such noises were not eliminated to provide a more realistic screening setting. The recording was performed on both children and adults. The recording was conducted while the subject was lying supine and in the left lateral decubitus position as shown in Figure 3. A total of 170 subjects had participated in the data collection process. One hundred twenty-four subjects had RHD, and 46 subjects were labeled as healthy control. The summary of the collected data is shown in Table 1. Only a single 30 s record from each subject was used in the analysis. Additional information about the dataset can be found in [52].

#### 2.1.2. Additional Heart Sound Dataset

Considering the latest incidence rate of RHD in Ethiopia (5.65%) [19], a screening tool deployed in the community will be exposed to more HC than PwRHD. However, the collected dataset had more heart sound data from PwRHD than from HC subjects.

To counter this inconsistency, additional records of HC were added from an open heart sound dataset [56,57], as shown in Table 1. This dataset contained 3126 heart sound recordings in total, collected from several subjects with a murmur or healthy controls. The data was compiled from several databases and labeled with the letters A through F. The given data had been contaminated by a variety of noise sources and that even in some of the recordings, it was impossible to distinguish between what was normal and what was abnormal in the data [57]. Each record was downsampled to 2 kHz and presented as a wav file. For the screening experiment, this study employed records from the healthy control group that were longer than 30 s and had just only one record per individual. Eighty-one records from 81 subjects satisfied these criteria.

### 2.2. Preprocessing

Each record was filtered with an antialiasing bandpass filter with a frequency band from 20 Hz to 1 kHz. All data were then downsampled to 2 kHz. Each recording was carefully labeled as RHD and Normal. Each recording was then divided into a 30-s duration, and one record from one subject was considered. The z-score of each signal was then computed to make a zero mean and unity standard deviation.

### 2.3. Feature Extraction

In machine learning, features greatly determine the performance of an algorithm. In this study, a unique set of features composed of time domain, frequency domain, perceptual domain, and acoustic domain were incorporated to improve the detection accuracy. A total of 31 features (acoustic domain (4), frequency domain (5), time domain (9), and perceptual domain (13)) were used to model the heart sound. Acoustic domain features have never been utilized in heart sound analysis systems before.

Time domain features include low and high-order statistics along with the energy content of the data in time domain representation. The frequency domain features contain features that indicate the energy content in the frequency domain as well as different sets of frequency values. The perceptual features model the physiological mechanisms of hearing, while the acoustic features relate the physical behavior of sound with the subjective response of sound [58]. Table 2 shows an overview of the complete set of extracted features. The features were computed using MATLAB2020a’s signal processing audio processing toolbox.

### 2.4. Classification

To classify the features as RHD or HC, a support vector machine (SVM) with radial basis function (RBF) kernel was used [59]. To quantitatively assess the generalization to previously unseen subjects, this paper proposes two nested cross-validation approaches. A brief description of SVM and nested cross-validation is presented below.

#### 2.4.1. SVMs

SVMs are powerful classifiers that can handle non-linear and non-separable classification tasks efficiently [60]. An SVM first maps the input data into a high-dimensional feature space and then searches for a separating hyperplane in this space that maximizes the separation between two classes. The primary idea is to select the optimal hyperplanes between distinct classes of data points [61]. For a given training data $\left(x_{i},\;y_{i}\right)$ where i=1 …N, SVM’s optimization problem is formulated as Equation (1).
(1)$$\underset{\omega ,\zeta ,b}{\min}J\left(\omega ,\zeta \right)=\frac{1}{2}\omega^{T}\omega +C\sum_{i=1}^{N}\zeta_{i},$$
such that
(2)$$y_{i}\left(\omega^{T}\varphi \left(x_{i}\right)+b\right)\geq 1-\zeta_{i},\;\;\;\;\;\;\;\;\;\;\;\;\;\;\;\;\;\;\;\;\;i=1,\;\ldots ,N$$
(3)$$\zeta_{i}\geq 0,\;\;\;\;\;\;\;\;\;\;\;\;\;\;\;\;\;\;\;\;\;\;\;\;\;\;\;\;\;\;\;\;\;\;\;i=1,\;\ldots ,N$$
where C  is an empirically chosen positive regularization constant, ω is the weight vector for training parameters, $\zeta_{i}$ is a positive slack variable that specifies the distance of $x_{i}$ to the decision boundary, and φ is a nonlinear mapping function used to map input data point $x_{i}$ into a higher dimensional space.

SVMs can be solved using Lagrange multipliers α ≥ 0 for Equation (4). The solution for the Lagrange multipliers is obtained by solving a quadratic programming problem. The SVM decision function can be expressed as Equation (4):(4)$$g\left(x\right)=\sum_{x_{i}\in SV}\alpha_{i}y_{i}K\left(x,x_{i}\right)+b$$
where $K\left(x,x_{i}\right)$ is the kernel function and defined as Equation (5):(5)$$K\left(x,x_{i}\right)=\varphi {\left(x\right)}^{T}\varphi \left(x_{i}\right)$$

In this paper, the Gaussian radial basis function was used. It is defined as (6):(6)$$K\left(x,x_{i}\right)=e^{-\gamma \|x-x_{i}\|{}^{2}}$$
and the SVM classifier was implemented using Scikit-learn version 0.21.3 [62] and Python version 3.6.8.

#### 2.4.2. Nested Cross-Validation Approach

To quantitatively assess the generalization to previously unseen subjects, this paper proposes two nested cross-validation approaches. In general, nested cross-validation partitions the dataset into an inner and outer cross-validation loop. In the inner loop, the hyperparameters are adjusted to optimize a model selection criterion. In the outer loop, the generalization of the optimized model to previously unseen subjects is assessed through a model evaluation criterion. This conservative protocol ensures complete separation among model selection and model evaluation.

The first nested cross-validation approach was termed stratified nested cross-validation. Stratified cross-validation preserves the proportion of classes across partitioning and thus forces each partition to be representative of the dataset. In the machine learning literature, stratification is used to reduce the internal variability of prediction error estimates [63]. For the second approach, the prevalence of RHD was set at 5%, while the proposed dataset contained a roughly equal number of PwRHD and HCs, at 124 and 127, respectively. However, in reality, the screening tool will be exposed to a larger proportion of HCs than present in the dataset. Given that hyperparameter optimization is sensitive to how the data is partitioned, the stratified nested cross-validation may result in sub-optimal hyperparameter choices [64]. Therefore, the model was additionally evaluated through a second nested cross-validation approach, termed imbalanced nested cross-validation, where the folds were partitioned according to a certain RHD prevalence. This approach is visualized in Figure 4. The imbalanced partitioning aims to mirror the clinically relevant use case where the RHD screening tool could be optimized for an a priori known RHD prevalence and then tasked to assess RHD in previously unseen subjects. 

Due to the repeated random sampling, the performance estimates were stochastic. Therefore, each experiment was repeated five times, resulting in five independent nested cross-validation experiments to assess the performance. For both cross-validation approaches, the model selection criterion optimized two hyperparameters; the flexibility of the decision boundary (complexity parameter), and the RBF kernel coefficient, denoted as C and γ, respectively. The hyperparameter grids were defined as: C = (0.01, 0.1, 1, 10, 100), and γ = (1, 0.1, 0.01, 0.001, 0.0001). The optimization aimed to maximize the f1-score across the inner testing partitions. The f1-score is defined as in Equation (9). The other objective metrics for performance evaluation were precision Equation (7), recall Equation (8), and specificity Equation (10).
(7)$$precision=\frac{T_{p}}{T_{p}+F_{p}}$$
(8)$$recall=\frac{T_{p}}{T_{p}+F_{N}}$$
(9)$$f1-score=2*\frac{precision*recall}{precision+recall\;}\;$$
(10)$$specificity=\frac{T_{N}}{T_{N}+F_{p}}$$
where *T_P_* = number of true positives, *F_P_* = number of false positives, *F_N_* = number of false negatives, and *T_N_* = number of true negatives.

## 3. Results

The proposed method was quantitively evaluated for RHD prevalence rates of 2.5%, 5%, 10%, 20%, and 10-fold stratified cross-validation. The results indicated that the f1-score improved as the prevalence rate increased. More specifically, the f1-score ranged from 59.0 ± 1.5% to 81.1 ± 1.5% for prevalence rates of 2.5% and 20%, respectively. A similar pattern was observed for the precision, which ranged from 44.2 ± 2.7% to 71.4 ± 2.0% for prevalence rates of 2.5% and 20%, respectively. However, the recall was consistently high independent of RHD prevalence. The other parameter which was consistently high throughout the different prevalence rates was specificity with a maximum value of 96.0 ± 0.6 achieved at 10-fold cross-validation. The highest f1-score of 96.0 ± 0.9% was also observed for the stratified 10-fold cross-validation experiment, where the prevalence of RHD and HCs were roughly equal across the partitions. All the results of this experiment are summarized in Table 3.

The second experiment was conducted to perform an objective comparison of different features used in six state-of-the-art heart sound classification algorithms. Features were extracted exactly as they are used in their respective papers. Then, the performance was evaluated in our RHD dataset. The f1-score, recall, precision, and specificity were computed for the 5% prevalence rate using nested cross-validation and for stratified-10-fold cross-validation. The result of this experiment is shown in Table 4. The utilization of the time domain feature alone, as proposed by Careena P. et al. [38], resulted in the lowest score. Using frequency domain [41] feature alone also had a lower classification performance. Compared to the time or frequency domain features, perceptual domain features alone [45] performed well. When perceptual features were combined with wavelet features [28], the classification performance was improved further. The best results were achieved by the different combinations of time, frequency, and perceptual features [52,53].

The final experiment was conducted with only the proposed time, frequency, and perceptual features, but without the acoustic features. This experiment aimed to assess the effect of the additional acoustic features on the classification performance which are new for the heart sound analysis. The same partitioning and evaluation procedure was followed to showcase the performance across different prevalence rates. More specifically, prevalence rates of 2.5%, 5%, 10%, and 20%. The f1-score was the lowest, at 2.5% (58.4 ± 1.4%), and increased as the prevalence rate increased. The highest score was 95.2 ± 0.8% under stratified 10-fold cross-validation. The recall increased from 85.5 ± 1.0% to 87.6 ± 2.6% when the prevalence rate increased from 2.5% to 5%. However, it dropped back at 10% and 20% with a value of 84.8 ± 2.0% and 83.9 ± 1.9%, respectively. The maximum recall of 96.1 ± 1.2% was recorded at stratified 10-fold cross-validation. Precision uniformly increased with increasing prevalence rate. Specificity was consistently high for all the prevalence rates, with its maximum being 95.2 ± 0.6% recorded at a 2.5% prevalence rate. The results of this experiment are summarized in Table 5.

## 4. Discussion

Classification of heart sound records from a phonocardiogram is challenging because the record is usually mixed with noise. This paper presented a screening tool to detect RHD for a possible community-level screening program. Extensive heart sound data from PwRHD and HC was collected for this purpose. The proposed approach was quantitively evaluated according to two nested cross-validation approaches to demonstrate the performance of the system against imbalanced data (Table 3). Table 4 includes a comparison of the proposed system with six sets of features utilized in state-of-the-art classification methods. An additional experiment was also conducted to demonstrate the effect of the proposed acoustics features on the classification performance (Table 5).

In the regular stratified nested 10-fold cross-validation, the model had an excellent detection performance with an f1-score of 96.0 ± 0.9%, a recall value of 95.8 ± 1.5%, a precision of 96.2 ± 0.6% and sensitivity of 96.0 ± 0.6%. In this stratified evaluation, the balanced class distribution of the dataset, i.e., 124 PwRHD and 127 HCs, was preserved across the hyperparameter and model evaluation partitions. In practice, however, such a balance in the data is not available.

As a detection tool, the system should be robust against a severe data imbalance common in a real-life scenario. In the Ethiopian context, the most recently reported population-wise prevalence rate of RHD was 5.67% [19]. To optimize and evaluate the model at prevalence rates that differ from the class distribution of the dataset, a second nested cross-validation approach, termed imbalanced nested cross-validation, was used. This approach partitioned model optimization and evaluation folds according to a certain a priori known RHD prevalence rate. An assessment of the influence of the prevalence rate on the hyperparameters optimization is provided in Appendix A (Figure A1). As the prevalence rate increases, the complexity parameter C decreases, whereas the kernel width increases. For low prevalence rates, the optimization proposes a higher cost of misclassification (a bigger C) and a smoother decision boundary (a smaller γ).

For a 5% prevalence rate of RHD, the model resulted in an f1-score of 72.2 ± 0.8%, a recall of 92.3 ± 0.4%, a precision of 59.2 ± 3.6% and a specificity of 94.8 ± 0.6%. In a detection tool, a high recall value is a vital requirement. That requirement has been fulfilled by our proposed method. A precision value of 59.2 ± 3.6% is relatively low, but this may be because the quality of the added open access data is low. The specificity at 5% prevalence was high 94.8 ± 0.6%, but the experiment always considered 1 PwRHD for 20 HC across all folds of the experiment; hence, the number of false positives will be a maximum of 1.

To further see the robustness of the model against restricted positive ratios, it was tested again for a prevalence of 2.5% of RHD. For this experiment, the system managed to detect most RHD cases with a recall value of 89.0 ± 1.2%. The precision was low at 44.2 ± 2.7%. To achieve the 2.5% prevalence rate, this experiment considered 1 PwRHD for 40 HC across all folds of the experiment. The proportion of samples in the evaluation partitions from the (lower quality) open-access dataset is even higher at a smaller prevalence rate. At a 2.5% prevalence rate, the number of HCs in the training partitions reduces as more HCs are preserved for evaluation. This increases the class imbalance in the training partitions and may in turn affect the capacity of the classifier to model the HCs.

During the systolic and diastolic stages of the heart cycle, RHD causes transient, non-stationary, high-frequency murmurs with low amplitude. It is critical to accurately describe abnormal cardiac murmurs to successfully detect them. As shown in Table 4, the use of purely temporal or spectral representations of the PCG as classification features is inadequate since they are limited in either temporal localization or frequency information which only achieved an f1-score of 83.2 ± 0.8% and 86.6 ± 1.2% for the stratified 10-fold cross-validation respectively. The modeling of heart sounds using MFCC to acquire perceptual information resulted in a reasonably good performance with an f1-score of 89.4 ± 1.5%. Improved results were obtained by combining time, frequency, and perceptual domain features with the highest f1-score in the group being 93.9 ± 0.4%. However, employing a unique and enhanced combination of time, frequency, perceptual and acoustic features had the best classification performance with an f1-score of 96.0 ± 0.9%. In all the evaluations metrics using the stratified 10-fold cross-validation, the performance of this article was better.

Further comparison was performed to observe the performance of each feature set using the 5% prevalence rate of the study area. In this case, the biggest f1-score of 72.2 ± 0.8% was achieved by our paper. However, the highest recall of 94.7 ± 1.6% was recorded by N. K Sawant et al. [53], but the corresponding precision and f1-score were low. Concerning the most important metrics in a hugely unbalanced dataset, the proposed approach performed best. This is because the proposed set of features comprise not only a combination of time, frequency, and perceptual features, but also acoustic features that model the sensations produced by sounds using distinct elements of sound pitch and intensity.

To see the effect of the proposed set of acoustic features, the proposed method was re-evaluated without these new sets of acoustic features (Table 5). It can be seen that there was a consistent performance improvement at all prevalence rates and stratified 10-fold cross-validation. A maximum of 10.7% increment of recall value was observed at a 10% prevalence rate. A 2.1% decrement of specificity at a 20% prevalence rate was also observed. In several cases, the two experiments produced almost identical results. Generally, acoustics features, which are common in audio and speech processing, can be also effectively utilized for heart sound processing. The observed higher performance of our approach indicates the proposed new and more elaborate set of features that incorporated attributes derived from time, frequency, perceptual, and acoustics domains have managed to effectively delineate murmur from normal heart sounds.

In the literature, automatic classification algorithms specifically developed for RHD detection were rare. Springer D. B. [28] reported an accuracy of 74.5 ± 2.1%, an f1-score of 69.9 ± 2.8%, a sensitivity of 74.8 ± 3.9%, and specificity of 74.1 ± 1.9%. Their data had over 36.7% prevalence rate of RHD. Even if our proposed approach and Springer D. B used different sets of datasets, both addressed the issue of RHD detection. It can be seen that our proposed method has a significantly improved recall value of 92.3 ± 0.4% and a better f1-score of 72.2 ± 0.8% at a lower prevalence rate of 5%. Springer D. B. revealed that an avoidable noise source had harmed the quality of their heart sound recordings gathered from RHD and normal participants. Conversely, their data, unlike ours, included recordings from borderline RHD cases where phonocardiogram may fail to detect.

Godown et al. [21] reported a sensitivity of 22.2% and a specificity of 91.2% of manual auscultation methods performed by experts at a prevalence rate of 3.4% definite RHD. To have a reasonable comparison, we ran our proposed algorithm at a 3.4% prevalence rate and an f1-score of 64.2 ± 3.7%: a recall value of 85.16 ± 1.7%, a precision of 51.6 ± 3.5% and a specificity of 95.2 ± 0.7% were achieved. When compared to manual auscultation, our suggested technique has a higher RHD sensitivity of 85.16 ± 1.7% and a better specificity of 95.2 ± 0.7%. Conversely, the reported sensitivity of 97.8% from low-cost handheld echocardiography was better to our proposed method. However, this high sensitivity was achieved by involving expert cardiologists that are not readily available in developing countries.

Our proposed method has a comparable result of detecting RHD when it was compared with non-experts who utilized low-cost handheld echocardiography for the detection of RHD. Ploutz M. et al. [23] reported sensitivity for definite RHD detection to be 90.9%. Our reported sensitivity was 89.0 ± 1.2% at the lowest 2.5% prevalence rate. Which is relatively identical to the performance of non-experts using low-cost handheld echocardiography. This indicates that there is a potential for automatic classifiers based on phonocardiogram to be utilized in RHD detection tasks in areas where advanced imaging devices such as echocardiography and expert cardiologists are not readily available.

## 5. Conclusions

RHD is a neglected disease that requires substantial effort to eradicate in developing nations where it is endemic. Transthoracic echocardiogram imaging is the primary method for RHD diagnosis and detection. which necessitates expert knowledge and is extremely expensive. Such privileges are not commonly available in countries where RHD is most prevalent. The effectiveness of the other cheapest alternative, namely cardiac auscultation, on the other hand, has declined over time and is difficult to master. Cheaper handheld echocardiograms have also been proposed. However, these devices still require specialized skills to perform RHD screening.

Task-shifting non-medical professionals to perform duties typically performed by physicians is one potential solution to limiting healthcare access. Screening for RHD qualifies as a task that can be easily transferred to a non-medically trained professional. As a consequence, in areas where RHD is prevalent, an automated heart sound analysis system could be important. This paper has demonstrated the capability of detecting RHD using such automated systems. It has shown better performance in objective metrics. The model has demonstrated a commendable RHD detection performance indicated by a recall value consistently higher than 90% at different prevalence rates. In the experiments, there were few missed RHD, which is important for a quick screening tool.

The results showed that the proposed method has a great potential to be utilized to detect RHD in children and young people with an f1-score of 72.2 ± 0.8%, a recall of 92.3 ± 0.4, a precision of 59.2 ± 3.6% and a specificity of 94.8 ± 0.6%. Furthermore, recall value of 89.0 ± 1.2%. was observed at low prevalence rates of RHD 2.5%. When compared to other state-of-the-art articles, our results consistently outperformed them because we were able to effectively predict murmur due mainly to RHD utilizing the proposed unique set of features.

However, for this paper, three main limitations are present. The model’s precision at low prevalence rates is the first and most obvious limitation; this can be improved by collecting more data and retraining the system. The second limitation is analogous to the procedure for obtaining imbalanced nested cross-validation. To mirror low RHD prevalence rates in the model optimization and evaluation partitions, HC subjects were randomly pooled from the inner and outer datasets. This results in overlap of HCs across partitions and thus reduced variety of that class. The third limitation is on the data, RHD records were obtained from patients seeking treatment in cardiac clinics who may have advanced stages of RHD.

In conclusion, the first significant contribution of this paper is the presentation of a carefully collected and standardized rheumatic heart disease heart sound data set. This has never been executed before. Second, it proposed a unique set of features capable of modeling heart sounds for improved detection capability. Third, rigorous evaluation of RHD detection at various prevalence rates, with the most critical parameter, recall, continuously above 90%, is critical for a screening tool.

Deploying such a system at school-level screening programs will have potentially important results mainly because it will enable mass screening and early detection. Furthermore, younger subjects with an early-stage diagnosis are the most to gain from secondary prophylaxis [65]. Finally, a non-medically trained person at school or in the community may use such a low-cost diagnostic method to screen those at high RHD risk and refer children with RHD for further confirmation with echocardiographic imaging and additional consultation with a medical doctor for further treatment. This effectively shifts the routine screening process away from the medical facility and physicians and toward the community and non-medically trained individuals. As a result, medical practitioners will be able to focus on their primary function, which is to treat patients.

## Figures and Tables

**Figure 1 sensors-21-06558-f001:**
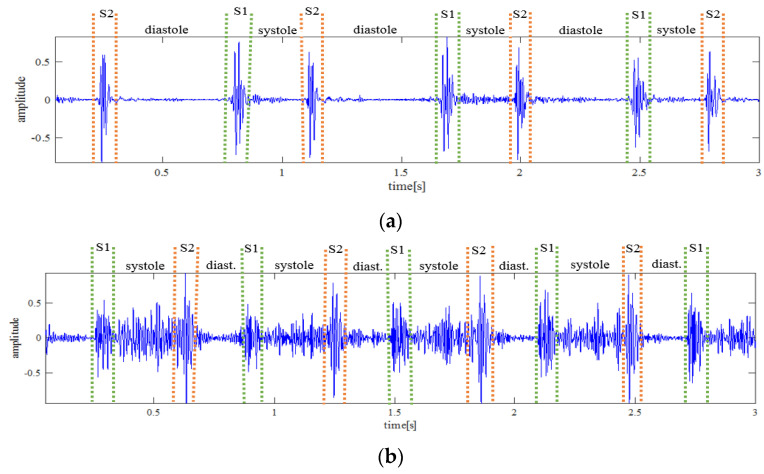
Graphical visualization of heart sound from: (**a**) recording from a HC with manual labeling of the fundamental heart sounds, and (**b**) recording from a PwRHD with audible murmur.

**Figure 2 sensors-21-06558-f002:**
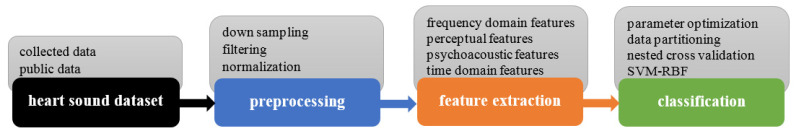
Block diagram for the overall methodology followed. It consists of four major blocks.

**Figure 3 sensors-21-06558-f003:**
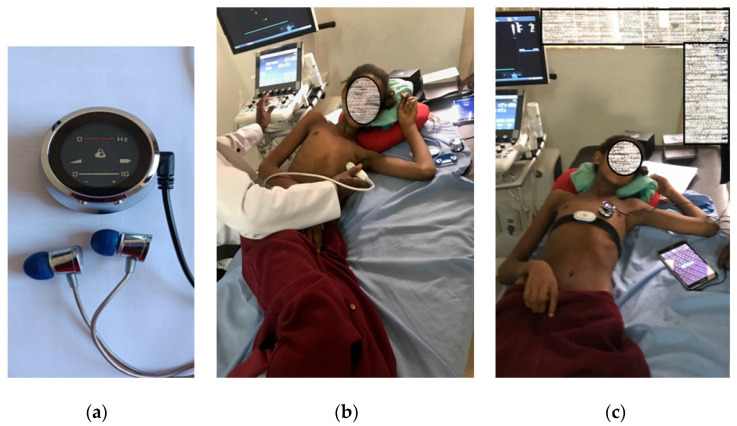
The data collection setup. (**a**) Thinklabs One™ electronic stethoscope, Thinklabs Medical LLC, CO, USA (**b**) transthoracic echocardiographic confirmation, and (**c**) the phonocardiogram recording.

**Figure 4 sensors-21-06558-f004:**
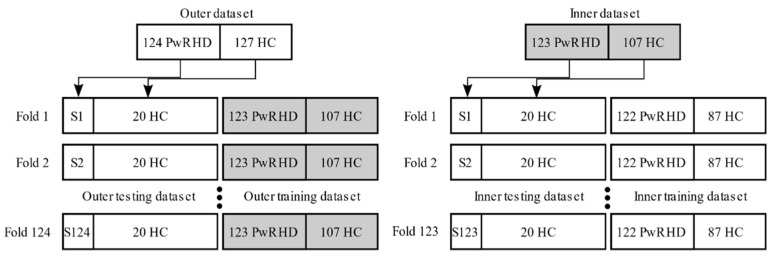
A visual overview of the imbalanced nested cross-validation approach. The nested approach used an outer loop to run a model evaluation criterion and an inner loop to optimize a model selection criterion. The outer and inner datasets were partitioned according to a leave-one RHD subject out approach. Given n PwRHD, the proposed partitioning results in n outer folds and n−1 inner folds. For each outer fold, the number of HCs to reach a certain desired RHD prevalence were randomly pooled from the outer dataset, i.e., 20 HCs to evaluate for the desired RHD prevalence of 5%. Similarly, for each inner fold, the number of HCs were randomly pooled from the inner dataset to optimize the hyperparameters for a certain desired RHD prevalence. This conservative approach ensured complete separation between model selection and model evaluation.

**Table 1 sensors-21-06558-t001:** Summary of the Ethiopian heart sound dataset collected from 170 subjects (~670 min of recordings), TASH, Addis Ababa, Ethiopia (row 1 and 2), and additional HC data from Physionet heart sound dataset (row 3–4), consisting of 81 normal subjects (~45 min of recordings).

No.	Dataset	No. of Records	Female/Male	Age (Years)	Average Recording Duration
1	Eth_PwRHD	124	74/50	22.9 ± 8.9	3.59 min
2	Eth_HC	46	15/31	14.4 ± 9.4	4.88 min
3	Physionet_cHC	3	* n.a.	29 ± 8	47 ± 25 s
4	Physionet_fHC	78	n.a.	56 ± 16	33 ± 5 s

* n.a., not available.

**Table 2 sensors-21-06558-t002:** Summary of features used in this study, grouped into four major parts as time domain, frequency domain, perceptual domain, and acoustic domain.

No.	Feature Group	Number of Features	List of Features
1	Acoustic domain	4	Acoustic Roughness
			Acoustic Loudness
			Acoustic Sharpness
			Acoustic Fluctuation
2	Frequency domain	5	Spectral entropy
			Dominant frequency value
			Dominant frequency magnitude
			Dominant frequency ratio
			Bandwidth
3	Time domain	9	Median
			Mean absolute deviation
			The first quartile
			The third quartile
			Interquartile range
			Skewness
			Kurtosis
			Shannon’s energy
			Zero Crossing rate
4	Perceptual domain	13	MFCC1 to MFCC13

**Table 3 sensors-21-06558-t003:** Results (mean ± standard deviation %) of the imbalanced nested cross-validation at four different prevalence rates and stratified nested 10-fold Cross-validation using the complete feature set.

Parameter	2.5%	5%	10%	20%	Stratified 10-Fold
f1-score	59.0 ± 1.7	72.2 ± 0.8	80.1 ± 1.5	81.1 ± 1.5	96.0 ± 0.9
Recall	89.0 ± 1.2	92.3 ± 0.4	95.6 ± 1.0	93.9 ± 1.2	95.8 ± 1.5
Precision	44.2 ± 2.7	59.2 ± 3.6	68.9 ± 3.3	71.4 ± 2.0	96.2 ± 0.6
specificity	95.2 ± 0.5	94.8 ± 0.6	92.6 ± 0.8	86.9 ± 1.8	96.0 ± 0.6

**Table 4 sensors-21-06558-t004:** Results (mean ± standard deviation %) of stratified nested 10-fold cross-validation and imbalanced nested cross-validation at 5% RHD prevalence rate using six sets of state-of-the-art features when evaluated on our RHD dataset.

Authors	Features (Numbers)	Evaluation	f1-Score	Recall	Precision	Specificity
Springer et al. [28]	Combination of undecimated wavelet transform (360) and MFCC (13)	10-fold CV	90.3 ± 2.0	86.3 ± 3.1	94.7 ± 0.9	94.6 ± 0.9
		nested CV at 5%	63.3 ± 2.4	72.4 ± 1.4	56.2 ± 3.0	94.3 ± 0.2
Careena P. et al. [38]	Time domain features (10)	10-fold CV	83.2 ± 0.8	82.0 ± 1.6	84.6 ± 1.7	84.4 ± 2.5
		nested CV at 5%	0.40 ± 1.6	67.9 ± 2.4	28.3 ± 1.5	90.4 ± 0.5
A. M. Alqudah et al. [41]	Frequency domain features (8)	10-fold CV	86.6 ± 1.2	89.7 ± 2.4	83.9 ± 3.6	82.1 ± 5.0
		nested CV at 5%	37.5 ± 1.0	85.8 ± 1.7	24.0 ± 0.9	82.9 ± 0.6
M. Deng et al. [45]	perceptual features (MFCC and its first and second derivatives) (26)	10-fold CV	89.4 ± 1.5	90.4 ± 1.9	88.5 ± 1.6	87.5 ± 1.9
		nested CV at 5%	44.6 ± 1.0	90.6 ± 1.5	29.6 ± 0.7	86.1 ± 0.2
Asmare et al. [52]	Combination of time (6), frequency (3) and perceptual features (13)	10-fold CV	93.9 ± 0.4	94.1 ± 1.1	93.7 ± 0.8	93.3 ± 0.9
		nested CV at 5%	66.8 ± 2.9	91.5 ± 1.9	52.6 ± 3.3	93.5 ± 0.7
N. K Sawant et al. [53]	Combination of time (3), frequency (4), and perceptual features (13)	10-fold CV	91.7 ± 1.3	90.8 ± 1.5	92.6 ± 1.2	92.6 ± 1.0
		nested CV at 5%	62.4 ± 2.2	**94.7 ± 1.6**	46.6 ± 2.3	91.7 ± 0.7
This paper	Combination of time (9), frequency (4), perceptual (13) and acoustic features (4)	10-fold CV	**96.0 ± 0.9**	**95.8 ± 1.5**	**96.2 ± 0.6**	**96.0 ± 0.6**
		nested CV at 5%	**72.2 ± 0.8**	92.3 ± 0.4	**59.2 ± 3.6**	**94.8 ± 0.6**

**Table 5 sensors-21-06558-t005:** Results (mean ± standard deviation %) of the imbalanced nested cross-validation at four different prevalence rates and stratified nested 10-fold cross-validation using only the time, frequency, and MFCC features.

Parameter	2.5%	5%	10%	20%	Stratified 10-Fold
f1-score	58.4 ± 1.4	68.7 ± 3.0	70.4 ± 2.1	73.8 ± 2.4	95.2 ± 0.8
Recall	85.5 ± 1.0	87.6 ± 2.6	84.8 ± 2.0	83.9 ± 1.9	96.1 ± 1.2
Precision	44.4 ± 2.5	56.5 ± 3.4	60.2 ± 2.2	65.9 ± 3.4	94.3 ± 0.7
specificity	95.2 ± 0.6	94.3 ± 0.5	92.1 ± 0.7	89.0 ± 1.5	93.7 ± 0.6

## Data Availability

Part of the data presented in this study are openly available in the Physionet repository. The ETH-RHD data that support the findings of this study are available on request from the corresponding author, M.H.A. The data are not publicly available due to intellectual property restrictions.

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
