# Peer review of "Rheumatic Heart Disease Screening Based on Phonocardiogram"

_sensors, 2021, doi:10.3390/s21196558_

Round 1
Reviewer 1 Report
This paper proposes a new and robust RHD screening tool from phonocardiogram recordings. The authors proposed an extensive RHD heart sound dataset, developed a screening tool by using comprehensive features.
However, there are two main problems:
(1) the classifier used is SVM, which is a classic but old method, why not try the current relatively new and popular classifier, such as deep learning?
(2) the experimental part only tests the method proposed in this paper, the comparison with other similar methods may be more convincing?
Author Response
Revision 01
Editor and reviewer comments
2021-09-19
Dear Editor,
Dear Reviewers,
We wish to thank the editor and the reviewers for your valuable time and for allowing us to submit a revision of our manuscript, entitled ‘Rheumatic Heart Disease Screening based on Phonocardiogram’ in MDPI Sensors.
We considered the many thoughtful and constructive suggestions raised by the reviewers. We tried to address each of the comments. All modifications to the original manuscript have been marked, as part of the “manuscript with track changes” file. All modifications have been made with respect to the included single-column version of the manuscript with line numbers along the margin.
We hope that the editor and the reviewers will consider this revision for publication in MDPI Sensors, and we look forward to reading your comments and decision. We are glad to address any further questions or comments.
On behalf of all co-authors,
Sincerely,
Melkamu Hunegnaw Asmare, MSc
Department of Electrical Engineering (ESAT)
KU Leuven
Reviewer #1:
Dear reviewer #1, thank you very much for the detailed review of the paper. The quality of the paper has now greatly improved. We used grammar checking tools to examine the writing and had the co-authors proofread it to improve the English.
Comments and Suggestions for Authors
This paper proposes a new and robust RHD screening tool from phonocardiogram recordings. The authors proposed an extensive RHD heart sound dataset, developed a screening tool by using comprehensive features. However, there are two main problems:
(1) the classifier used is SVM, which is a classic but old method, why not try the current relatively new and popular classifier, such as deep learning?
Author Response: You are right. Over the past few years, various generic deep learning (DL) approaches have been proposed that present interesting architectural characteristics for modeling heart sounds [1] [2]. However, the objective of the present study was to detect RHD for screening. Due to the relatively low amount of healthy control subjects compared to RHD subjects in the present dataset, we set up a computationally expensive imbalanced cross-validation protocol to assess screening performance. Unfortunately, it is computationally infeasible to evaluate and optimize DL models under the proposed protocol. For reference, a single training iteration through our dataset of the WaveNet DL architecture proposed in [1] takes roughly 190 seconds on an NVIDIA Tesla K80. In comparison, a training iteration for SVM with the proposed feature set takes 0.005 seconds. Considering that the entire imbalanced (nested) cross-validation protocol can take up to 8 hours for optimizing and evaluating the SVM, the DL approach would be computationally infeasible. Therefore, DL is out of the scope of the present study and we instead focussed on feature-based approaches. After acquiring more data from a healthy control population and thereby removing the computational constraint of the current protocol, we agree that it would be interesting to compare DL approaches in the future for screening RHD.
[1] Oh, Shu Lih, V. Jahmunah, Chui Ping Ooi, Ru-San Tan, Edward J. Ciaccio, Toshitaka Yamakawa, Masayuki Tanabe, Makiko Kobayashi, and U. Rajendra Acharya. 2020. “Classification of Heart Sound Signals Using a Novel Deep WaveNet Model.” Computer Methods and Programs in Biomedicine 196 (November): 105604.​
[2] Latif, Siddique, Muhammad Usman, Rajib Rana, and Junaid Qadir. 2018. “Phonocardiographic Sensing Using Deep Learning for Abnormal Heartbeat Detection.” IEEE Sensors Journal 18 (22): 9393–9400.
(2) the experimental part only tests the method proposed in this paper, the comparison with other similar methods may be more convincing?
Author Response: Yes, you are right. To provide additional context to the achieved results, we have now included a comparative study with some of the latest reported heart sound classification features. Six feature-based heart sound classification approaches were considered for the comparative study. To make the comparisons, we computed the features as explained in each respective article and applied them to our proposed classification algorithm and on our RHD dataset. The performance of these algorithms was compared using the stratified 10-fold cross-validation. To further assess the performance we also evaluated them using the nested-cross validation at 5% RHD prevalence which is the reported prevalence rate.
Author Action: We updated the manuscript by adding more background information about the articles which are selected for the comparative study which is found in the Introduction Section in lines 173-217.
Careena P. et al. [38] extracted variance, standard deviation, entropy, peak amplitude, RMS, crest factor, impulse factor, shape factor, energy, and clearance factor as time-domain features to classify heart sounds. They argued that time-domain features are analytically straightforward to compute. Their algorithm was evaluated on a public dataset and resulted in good classification performance.
M. Alqudah et al. [41] used instantaneous frequency estimation features where the instantaneous frequency of a non-stationary signal was defined as a time-varying parameter that is related to the mean of the frequencies in the signal as it develops and computed as the first conditional spectral moment of the power spectrum. They extracted 8 different frequency domain features to capture the instantaneous frequency properties of heart sound signals.
Deng et al.[45] capitalized on the Mel Frequency Cepstral Coefficient (MFCC) feature’s excellent performance in many sound perceptions and tried to improve the MFCC features to elaborate the dynamic characteristics among consecutive heart sound signals. They argued that, due to the higher sensitivity of the human ear to the dynamic characteristics of acoustic signals, the dynamic information contained in the heart sounds spectrum also offers a wealth of information that can be used to further improve classification accuracy. The first and second derivatives of the MFCC coefficients are also included as features to reflect the dynamic information included in the heart sound signal.
K Sawant et al. [55] have proposed time-domain features which include zero-crossing rate, the energy of the raw signal and entropy; frequency-domain features which include spectral centroid, spectral spread, spectral flux, and spectral entropy; perceptual features which incorporate the first 7 MFCC coefficients.
Springer et al. [28] proposed an automated detection of a specific heart murmur caused by RHD. They developed an SVM-based classification algorithm which was tested on an in-house dataset, collected from PwRHD and control groups in South Africa. They used undecimated wavelet transform (UWT) and Mel Frequency Cepstral Coefficients (MFCC) in their feature extraction stage. They claimed that UWT was selected over the other wavelet transforms as it provides a wavelet decomposition at discrete decomposition levels, leading to a reduction in the feature space, while not halving the number of samples in the signal at each decomposition level. They used the level 4 Daubechies 5 (db5) wavelet and extracted detail coefficients from level 2, level 3, and level 4. To reduce the feature space, they used the median of a specified window.
Asmare M. H. et al. [44], [54] have also proposed heart sound classification to detect murmur from RHD. They extracted 26 features from the entire heart sound signal to properly deal with systolic as well as diastolic murmurs caused by RHD [54]. Their features consist of the time domain, frequency domain, and perceptual domain.
The results of the comparative study are now presented in Table 4 and the following text is inserted in the article. Refer to lines 398-414 of the Results Section.
The second experiment was conducted to perform an objective comparison of different features used in different state-of-the-art heart sound classification algorithms. Features were extracted exactly as they are used in their respective papers. Then the performance was evaluated in our RHD dataset. The f1-score, recall, precision, and specificity were computed for the 5% prevalence rate using nested cross-validation and for stratified-10-fold cross-validation used. The result of this experiment is shown in Table 4. The utilization of the time-domain feature alone as proposed by Careena P. et al. [38] resulted in the lowest score. Using frequency domain [41] feature alone has also had a lower classification performance. Compared to the time or frequency domain features, perceptual domain features [45] performed better. When perceptual features were combined with wavelet features [28] the classification performance was improved further. Better results were achieved by the different combinations of time, frequency, and perceptual features were included [54], [55].
Table 4. Results (mean ± standard deviation %) of stratified nested 10-fold cross-validation and imbalanced nested cross-validation at 5% RHD prevalence rate using different sets of features as stated in six different articles.
Authors |
Features (numbers) |
Evaluation |
f1-score |
Recall |
Precision |
Specificity |
Springer et al. [28] |
Combination of undecimated wavelet transform (360) and MFCC(13) |
10-fold CV |
90.3±2.0 |
86.3±3.1 |
94.7±0.9 |
94.6±0.9 |
nested CV at 5% |
63.3±2.4 |
72.4±1.4 |
56.2±3.0 |
94.3±0.2 |
||
Careena P. et al. [38] |
Time-domain features(10) |
10-fold CV |
83.2±0.8 |
82.0±1.6 |
84.6±1.7 |
84.4±2.5 |
nested CV at 5% |
0.40±1.6 |
67.9±2.4 |
28.3±1.5 |
90.4±0.5 |
||
A. M. Alqudah et al. [41] |
Frequency domain features(8) |
10-fold CV |
86.6±1.2 |
89.7±2.4 |
83.9±3.6 |
82.1±5.0 |
nested CV at 5% |
37.5±1.0 |
85.8±1.7 |
24.0±0.9 |
82.9±0.6 |
||
M. Deng et al. [45] |
perceptual features(MFCC and its first and second derivatives) (26) |
10-fold CV |
89.4±1.5 |
90.4±1.9 |
88.5±1.6 |
87.5±1.9 |
nested CV at 5% |
44.6±1.0 |
90.6±1.5 |
29.6±0.7 |
86.1±0.2 |
||
Asmare et al. [54] |
Combination of time(6), frequency(3) and perceptual features(13) |
10-fold CV |
93.9±0.4 |
94.1±1.1 |
93.7±0.8 |
93.3±0.9 |
nested CV at 5% |
66.8±2.9 |
91.5±1.9 |
52.6±3.3 |
93.5±0.7 |
||
N. K Sawant et al. [55] |
Combination of time(3), frequency(4), and perceptual features(13) |
10-fold CV |
91.7±1.3 |
90.8±1.5 |
92.6±1.2 |
92.6±1.0 |
nested CV at 5% |
62.4±2.2 |
94.7±1.6 |
46.6±2.3 |
91.7±0.7 |
||
This paper |
Combination of time(9), frequency(4), perceptual(13) and acoustic features(4) |
10-fold CV |
96.0±0.9 |
95.8±1.5 |
96.2±0.6 |
96.0±0.6 |
nested CV at 5% |
72.2±0.8 |
92.3±0.4 |
59.2±3.6 |
94.8±0.6 |
These comparative study results are discussed in the Discussion Section and the following text is inserted in lines 476-498.
During the systolic and diastolic stages of the heart cycle, RHD causes transient, non-stationary, high-frequency murmurs with low amplitude. It is critical to accurately describe abnormal cardiac murmurs to successfully detect them. As shown in Table 4, the use of purely temporal or spectral representations of the PCG as classification features was inadequate since they are limited in either temporal localization or frequency information which only achieved an f1-score of 83.2±0.8% and 86.6±1.2% for the stratified 10-fold cross-validation respectively. The modeling of heart sounds using MFCC to acquire perceptual information resulted in a reasonably good performance with an f1-score of 89.4±1.5%. Improved results were obtained by combining time, frequency, and perceptual domain features with the highest f1-score in the group being 93.9±0.4%. However, employing a unique and enhanced combination of time, frequency, perceptual and acoustic features had the best classification performance with an f1-score of 96.0±0.9%. In all the evaluations metrics using the stratified 10-fold cross-validation, the performance of this article was better.
Further comparison was done to see the performance of each feature set using the 5% prevalence rate of the study area, In this case, the biggest f1-score of 72.2±0.8% was achieved by our paper. However, the highest recall of 94.7±1.6% was recorded by N. K Sawant et al. [55] but the corresponding precision and f1-score were low. Concerning the most important metrics in a hugely unbalanced dataset, the proposed approach performed best. This is because the proposed set of features comprise not only a combination of time, frequency, perceptual features but also acoustic features which model the sensations produced by sounds using distinct elements of sound pitch and intensity.
And finally, the following text is inserted in the conclusion section. Refer to lines 559-562.
When compared to other state-of-the-art articles, our results consistently outperformed them because we were able to effectively predict murmur due mainly to RHD utilizing the proposed unique set of features.
Reviewer 2 Report
The paper is well written and the results are every interesting. I have two minor questions. 1. What is the cost of the device shown in Fig.3, and this could prevent its widespread usage? 2. It is better to have a benchmarking towards similar work. Probably it is difficult to find, but still comparing with similar method would be very interesting to read.
Author Response
Revision 01
Editor and reviewer comments
2021-09-19
Dear Editor,
Dear Reviewer,
We wish to thank the editor and the reviewers for your valuable time and for allowing us to submit a revision of our manuscript, entitled ‘Rheumatic Heart Disease Screening based on Phonocardiogram’ in MDPI Sensors.
We considered the many thoughtful and constructive suggestions raised by the reviewers. We tried to address each of the comments. All modifications to the original manuscript have been marked, as part of the “manuscript with track changes” file. All modifications have been made with respect to the included single-column version of the manuscript with line numbers along the margin.
We hope that the editor and the reviewers will consider this revision for publication in MDPI Sensors, and we look forward to reading your comments and decision. We are glad to address any further questions or comments.
On behalf of all co-authors,
Sincerely,
Melkamu Hunegnaw Asmare, MSc
Department of Electrical Engineering (ESAT)
KU Leuven
Reviewer #2:
Comments and Suggestions for Authors
The paper is well written and the results are very interesting. I have two minor questions.
Dear reviewer #2, thank you very much for the detailed review of the paper. The quality of the paper has now greatly improved. We used grammar checking tools to examine the writing and had the co-authors proofread it to improve the English.
Comments and Suggestions for Authors
- What is the cost of the device shown in Fig.3, and this could prevent its widespread usage?
Author Response: The cost of the ThinkLabs One ™ electronic stethoscope was 499USD. You are right, the price is high compared to an analog stethoscope. However, it is still considerably cheaper than a handheld echocardiology machine (for example ACUSON P10™) which costs over 5000USD and still requires an expert physician to use it.
But most importantly, in developing countries, the number of highly trained physicians like cardiologists is very low which makes medical care more expensive. This approach, when deployed can be used by a non-medically trained individual to screen RHD which effectively task shifts the routine screening process from the medical facility and physicians to the community and non-medically trained people. In doing so, medical professionals will be freed up to do their main job, i.e. treating people. On top of that, we have had a preliminary discussion with the device manufacturers to collaborate further to valorize our idea where we hope the cost will further reduce.
Author Action: The following text is now inserted in the Conclusion section lines 584-587.
This effectively shifts the routine screening process away from the medical facility and physicians and toward the community and non-medically trained individuals. As a result, medical practitioners will be able to focus on their primary function, which is to treat patients.
- It is better to have a benchmarking towards similar work. Probably it is difficult to find, but still comparing with similar method would be very interesting to read.
Author Response: Yes, you are right. To provide additional context to the achieved results, we have now included a comparative study with some of the latest reported heart sound classification features. Six feature-based heart sound classification approaches were considered for the comparative study. To make the comparisons, we computed the features as explained in each respective article and applied them to our proposed classification algorithm and on our RHD dataset. The performance of these algorithms was compared using the stratified 10-fold cross-validation. To further assess the performance we also evaluated them using the nested-cross validation at 5% RHD prevalence which is the reported prevalence rate.
Author Action: We updated the manuscript by adding more background information about the articles which are selected for the comparative study which is found in the Introduction Section in lines 173-217.
Careena P. et al. [38] extracted variance, standard deviation, entropy, peak amplitude, RMS, crest factor, impulse factor, shape factor, energy, and clearance factor as time-domain features to classify heart sounds. They argued that time-domain features are analytically straightforward to compute. Their algorithm was evaluated on a public dataset and resulted in good classification performance.
M. Alqudah et al. [41] used instantaneous frequency estimation features where the instantaneous frequency of a non-stationary signal was defined as a time-varying parameter that is related to the mean of the frequencies in the signal as it develops and computed as the first conditional spectral moment of the power spectrum. They extracted 8 different frequency domain features to capture the instantaneous frequency properties of heart sound signals.
Deng et al.[45] capitalized on the Mel Frequency Cepstral Coefficient (MFCC) feature’s excellent performance in many sound perceptions and tried to improve the MFCC features to elaborate the dynamic characteristics among consecutive heart sound signals. They argued that, due to the higher sensitivity of the human ear to the dynamic characteristics of acoustic signals, the dynamic information contained in the heart sounds spectrum also offers a wealth of information that can be used to further improve classification accuracy. The first and second derivatives of the MFCC coefficients are also included as features to reflect the dynamic information included in the heart sound signal.
K Sawant et al. [55] have proposed time-domain features which include zero-crossing rate, the energy of the raw signal and entropy; frequency-domain features which include spectral centroid, spectral spread, spectral flux, and spectral entropy; perceptual features which incorporate the first 7 MFCC coefficients.
Springer et al. [28] proposed an automated detection of a specific heart murmur caused by RHD. They developed an SVM-based classification algorithm which was tested on an in-house dataset, collected from PwRHD and control groups in South Africa. They used undecimated wavelet transform (UWT) and Mel Frequency Cepstral Coefficients (MFCC) in their feature extraction stage. They claimed that UWT was selected over the other wavelet transforms as it provides a wavelet decomposition at discrete decomposition levels, leading to a reduction in the feature space, while not halving the number of samples in the signal at each decomposition level. They used the level 4 Daubechies 5 (db5) wavelet and extracted detail coefficients from level 2, level 3, and level 4. To reduce the feature space, they used the median of a specified window.
Asmare M. H. et al. [44], [54] have also proposed heart sound classification to detect murmur from RHD. They extracted 26 features from the entire heart sound signal to properly deal with systolic as well as diastolic murmurs caused by RHD [54]. Their features consist of the time domain, frequency domain, and perceptual domain.
The results of the comparative study are now presented in Table 4 and the text in the following text is inserted in the article. Refer to lines 398-414 of the Results Section.
The second experiment was conducted to perform an objective comparison of different features used in different state-of-the-art heart sound classification algorithms. Features were extracted exactly as they are used in their respective papers. Then the performance was evaluated in our RHD dataset. The f1-score, recall, precision, and specificity were computed for the 5% prevalence rate using nested cross-validation and for stratified-10-fold cross-validation used. The result of this experiment is shown in Table 4. The utilization of the time-domain feature alone as proposed by Careena P. et al. [38] resulted in the lowest score. Using frequency domain [41] feature alone has also had a lower classification performance. Compared to the time or frequency domain features, perceptual domain features [45] performed better. When perceptual features were combined with wavelet features [28] the classification performance was improved further. Better results were achieved by the different combinations of time, frequency, and perceptual features were included [54], [55].
Table 4. Results (mean ± standard deviation %) of stratified nested 10-fold cross-validation and imbalanced nested cross-validation at 5% RHD prevalence rate using different sets of features as stated in six different articles.
Authors |
Features (numbers) |
Evaluation |
f1-score |
Recall |
Precision |
Specificity |
Springer et al. [28] |
Combination of undecimated wavelet transform (360) and MFCC(13) |
10-fold CV |
90.3±2.0 |
86.3±3.1 |
94.7±0.9 |
94.6±0.9 |
nested CV at 5% |
63.3±2.4 |
72.4±1.4 |
56.2±3.0 |
94.3±0.2 |
||
Careena P. et al. [38] |
Time-domain features(10) |
10-fold CV |
83.2±0.8 |
82.0±1.6 |
84.6±1.7 |
84.4±2.5 |
nested CV at 5% |
0.40±1.6 |
67.9±2.4 |
28.3±1.5 |
90.4±0.5 |
||
A. M. Alqudah et al. [41] |
Frequency domain features(8) |
10-fold CV |
86.6±1.2 |
89.7±2.4 |
83.9±3.6 |
82.1±5.0 |
nested CV at 5% |
37.5±1.0 |
85.8±1.7 |
24.0±0.9 |
82.9±0.6 |
||
M. Deng et al. [45] |
perceptual features(MFCC and its first and second derivatives) (26) |
10-fold CV |
89.4±1.5 |
90.4±1.9 |
88.5±1.6 |
87.5±1.9 |
nested CV at 5% |
44.6±1.0 |
90.6±1.5 |
29.6±0.7 |
86.1±0.2 |
||
Asmare et al. [54] |
Combination of time(6), frequency(3) and perceptual features(13) |
10-fold CV |
93.9±0.4 |
94.1±1.1 |
93.7±0.8 |
93.3±0.9 |
nested CV at 5% |
66.8±2.9 |
91.5±1.9 |
52.6±3.3 |
93.5±0.7 |
||
N. K Sawant et al. [55] |
Combination of time(3), frequency(4), and perceptual features(13) |
10-fold CV |
91.7±1.3 |
90.8±1.5 |
92.6±1.2 |
92.6±1.0 |
nested CV at 5% |
62.4±2.2 |
94.7±1.6 |
46.6±2.3 |
91.7±0.7 |
||
This paper |
Combination of time(9), frequency(4), perceptual(13) and acoustic features(4) |
10-fold CV |
96.0±0.9 |
95.8±1.5 |
96.2±0.6 |
96.0±0.6 |
nested CV at 5% |
72.2±0.8 |
92.3±0.4 |
59.2±3.6 |
94.8±0.6 |
These comparative study results are discussed in the Discussion Section and the following text is inserted in lines 476-498.
During the systolic and diastolic stages of the heart cycle, RHD causes transient, non-stationary, high-frequency murmurs with low amplitude. It is critical to accurately describe abnormal cardiac murmurs to successfully detect them. As shown in Table 4, the use of purely temporal or spectral representations of the PCG as classification features was inadequate since they are limited in either temporal localization or frequency information which only achieved an f1-score of 83.2±0.8% and 86.6±1.2% for the stratified 10-fold cross-validation respectively. The modeling of heart sounds using MFCC to acquire perceptual information resulted in a reasonably good performance with an f1-score of 89.4±1.5%. Improved results were obtained by combining time, frequency, and perceptual domain features with the highest f1-score in the group being 93.9±0.4%. However, employing a unique and enhanced combination of time, frequency, perceptual and acoustic features had the best classification performance with an f1-score of 96.0±0.9%. In all the evaluations metrics using the stratified 10-fold cross-validation, the performance of this article was better.
Further comparison was done to see the performance of each feature set using the 5% prevalence rate of the study area, In this case, the biggest f1-score of 72.2±0.8% was achieved by our paper. However, the highest recall of 94.7±1.6% was recorded by N. K Sawant et al. [55] but the corresponding precision and f1-score were low. Concerning the most important metrics in a hugely unbalanced dataset, the proposed approach performed best. This is because the proposed set of features comprise not only a combination of time, frequency, perceptual features but also acoustic features which model the sensations produced by sounds using distinct elements of sound pitch and intensity.
And finally, the following text is inserted in the conclusion section. Refer to lines 559-562.
When compared to other state-of-the-art articles, our results consistently outperformed them because we were able to effectively predict murmur due mainly to RHD utilizing the proposed unique set of features.
Reviewer 3 Report
This work studies the use of machine learning to detect the presence of rheumatic heart diseases (RHDs) from the analysis of phonocardiogram data. For this purpose, the authors use a specific "in-house" dataset, which was described in a previous work by the authors [42]. The proposed approach, is based on an SVM classifier and introduces the use of acoustic features for RHD heart sound detection. Experiments over the considered dataset show a significant potential of the proposed method in detecting the presence of RHD from heart sound analysis. Main comments:
- Although the paper considers a relevant problem, with significant societal impact, for which a specific large dataset is provided, the main concern with this work is related to the actual novelty of its technical contribution. In particular, the use of SVMs in conjunction with features extracted from heart sounds seems not novel. More specifically, the authors should compare the results obtained with the proposed method with those obtained by other approaches in the literature (e.g., the one described in [28]) on the same dataset and describe more clearly the novelty of their approach with respect to them. In fact, the paper claims to achieve "superior RHD detection capability" without a fair comparison. Comparisons with other classification techniques for RHD detection that were run over different datasets have very limited value.
- The authors should underline more explicitly what are the specific challenges associated with the detection of RHD compared to the more common task of abnormal heart sound detection, for which a large amount of work has already appeared in the literature. In particular, it is not clear if a machine learning method originally designed to perform abnormal heart sound detection could be adapted to the RHD detection problem, by simply training the algorithm on the proposed dataset.
Minor comments:
- In lines 110-111, the authors mention the fact that cheaper and more robust strategies in the detection of RHD than echocardiography are highly sought. However, it is not clear how auscultation could represent a more robust strategy for RHD detection than echocardiography.
- In line 138, "to have sensitivity and specificity" should be "to have low sensitivity and specificity"?
- In line 237 and in Table 1, there seems to be inconsistency in the number of normal subjects extracted from the PhysioNet dataset (81 vs. 111).
- In line 246, antialiasing filtering should be applied before downsampling, in order to avoid aliasing.
- The Lagrange multiplier alpha could be mentioned after equation (4) to avoid confusion.
- In line 295: "gaussian" -> "Gaussian".
- In line 327: "gamma (\gamma)" -> "\gamma".
- When considering highly imbalanced training datasets, it would be interesting to consider the use of specific oversampling techniques to reduce the effect of such imbalance.
Author Response
Revision 01
Editor and reviewer comments
2021-09-19
Dear Editor,
Dear Reviewer,
We wish to thank the editor and the reviewers for your valuable time and for allowing us to submit a revision of our manuscript, entitled ‘Rheumatic Heart Disease Screening based on Phonocardiogram’ in MDPI Sensors.
We considered the many thoughtful and constructive suggestions raised by the reviewers. We tried to address each of the comments. All modifications to the original manuscript have been marked, as part of the “manuscript with track changes” file. All modifications have been made with respect to the included single-column version of the manuscript with line numbers along the margin.
We hope that the editor and the reviewers will consider this revision for publication in MDPI Sensors, and we look forward to reading your comments and decision. We are glad to address any further questions or comments.
On behalf of all co-authors,
Sincerely,
Melkamu Hunegnaw Asmare, MSc
Department of Electrical Engineering (ESAT)
KU Leuven
Reviewer #3:
Dear reviewer #3, thank you very much for the detailed review of the paper. The quality of the paper has now greatly improved. We used grammar checking tools to examine the writing and had the co-authors proofread it to improve the English.
Comments and Suggestions for Authors
This work studies the use of machine learning to detect the presence of rheumatic heart diseases (RHDs) from the analysis of phonocardiogram data. For this purpose, the authors use a specific "in-house" dataset, which was described in a previous work by the authors [42]. The proposed approach, is based on an SVM classifier and introduces the use of acoustic features for RHD heart sound detection. Experiments over the considered dataset show a significant potential of the proposed method in detecting the presence of RHD from heart sound analysis.
Main comments:
- Although the paper considers a relevant problem, with significant societal impact, for which a specific large dataset is provided, the main concern with this work is related to the actual novelty of its technical contribution. In particular, the use of SVMs in conjunction with features extracted from heart sounds seems not novel. More specifically, the authors should compare the results obtained with the proposed method with those obtained by other approaches in the literature (e.g., the one described in [28]) on the same dataset and describe more clearly the novelty of their approach with respect to them. In fact, the paper claims to achieve "superior RHD detection capability" without a fair comparison. Comparisons with other classification techniques for RHD detection that were run over different datasets have very limited value.
Author Response: Yes, the use of SVMs in conjunction with features extracted from heart sounds is not novel. However, in this article, a more complete set of features were proposed. This article tried to solve the issue of screening which has not been attempted before. The claim "superior RHD detection capability" is replaced with "better RHD detection capability". To provide additional context to the achieved results, we have now included a comparative study with some of the latest reported heart sound classification features. Six feature-based heart sound classification approaches were considered for the comparative study. To make the comparisons, we computed the features as explained in each respective article and applied them to our proposed classification algorithm and on our RHD dataset. The performance of these algorithms was compared using the stratified 10-fold cross-validation. To further assess the performance we also evaluated them using the nested-cross validation at 5% RHD prevalence which is the reported prevalence rate.
Author Action: We updated the manuscript by adding more background information about the articles which are selected for the comparative study which is found in the Introduction Section in lines 173-217.
Careena P. et al. [38] extracted variance, standard deviation, entropy, peak amplitude, RMS, crest factor, impulse factor, shape factor, energy, and clearance factor as time-domain features to classify heart sounds. They argued that time-domain features are analytically straightforward to compute. Their algorithm was evaluated on a public dataset and resulted in good classification performance.
M. Alqudah et al. [41] used instantaneous frequency estimation features where the instantaneous frequency of a non-stationary signal was defined as a time-varying parameter that is related to the mean of the frequencies in the signal as it develops and computed as the first conditional spectral moment of the power spectrum. They extracted 8 different frequency domain features to capture the instantaneous frequency properties of heart sound signals.
Deng et al.[45] capitalized on the Mel Frequency Cepstral Coefficient (MFCC) feature’s excellent performance in many sound perceptions and tried to improve the MFCC features to elaborate the dynamic characteristics among consecutive heart sound signals. They argued that, due to the higher sensitivity of the human ear to the dynamic characteristics of acoustic signals, the dynamic information contained in the heart sounds spectrum also offers a wealth of information that can be used to further improve classification accuracy. The first and second derivatives of the MFCC coefficients are also included as features to reflect the dynamic information included in the heart sound signal.
K Sawant et al. [55] have proposed time-domain features which include zero-crossing rate, the energy of the raw signal and entropy; frequency-domain features which include spectral centroid, spectral spread, spectral flux, and spectral entropy; perceptual features which incorporate the first 7 MFCC coefficients.
Springer et al. [28] proposed an automated detection of a specific heart murmur caused by RHD. They developed an SVM-based classification algorithm which was tested on an in-house dataset, collected from PwRHD and control groups in South Africa. They used undecimated wavelet transform (UWT) and Mel Frequency Cepstral Coefficients (MFCC) in their feature extraction stage. They claimed that UWT was selected over the other wavelet transforms as it provides a wavelet decomposition at discrete decomposition levels, leading to a reduction in the feature space, while not halving the number of samples in the signal at each decomposition level. They used the level 4 Daubechies 5 (db5) wavelet and extracted detail coefficients from level 2, level 3, and level 4. To reduce the feature space, they used the median of a specified window.
Asmare M. H. et al. [44], [54] have also proposed heart sound classification to detect murmur from RHD. They extracted 26 features from the entire heart sound signal to properly deal with systolic as well as diastolic murmurs caused by RHD [54]. Their features consist of the time domain, frequency domain, and perceptual domain.
The results of the comparative study are now presented in Table 4 and the text in the following text is inserted in the article. Refer to lines 398-414 of the Results Section.
The second experiment was conducted to perform an objective comparison of different features used in different state-of-the-art heart sound classification algorithms. Features were extracted exactly as they are used in their respective papers. Then the performance was evaluated in our RHD dataset. The f1-score, recall, precision, and specificity were computed for the 5% prevalence rate using nested cross-validation and for stratified-10-fold cross-validation used. The result of this experiment is shown in Table 4. The utilization of the time-domain feature alone as proposed by Careena P. et al. [38] resulted in the lowest score. Using frequency domain [41] feature alone has also had a lower classification performance. Compared to the time or frequency domain features, perceptual domain features [45] performed better. When perceptual features were combined with wavelet features [28] the classification performance was improved further. Better results were achieved by the different combinations of time, frequency, and perceptual features were included [54], [55].
Table 4. Results (mean ± standard deviation %) of stratified nested 10-fold cross-validation and imbalanced nested cross-validation at 5% RHD prevalence rate using different sets of features as stated in six different articles.
Authors |
Features (numbers) |
Evaluation |
f1-score |
Recall |
Precision |
Specificity |
Springer et al. [28] |
Combination of undecimated wavelet transform (360) and MFCC(13) |
10-fold CV |
90.3±2.0 |
86.3±3.1 |
94.7±0.9 |
94.6±0.9 |
nested CV at 5% |
63.3±2.4 |
72.4±1.4 |
56.2±3.0 |
94.3±0.2 |
||
Careena P. et al. [38] |
Time-domain features(10) |
10-fold CV |
83.2±0.8 |
82.0±1.6 |
84.6±1.7 |
84.4±2.5 |
nested CV at 5% |
0.40±1.6 |
67.9±2.4 |
28.3±1.5 |
90.4±0.5 |
||
A. M. Alqudah et al. [41] |
Frequency domain features(8) |
10-fold CV |
86.6±1.2 |
89.7±2.4 |
83.9±3.6 |
82.1±5.0 |
nested CV at 5% |
37.5±1.0 |
85.8±1.7 |
24.0±0.9 |
82.9±0.6 |
||
M. Deng et al. [45] |
perceptual features(MFCC and its first and second derivatives) (26) |
10-fold CV |
89.4±1.5 |
90.4±1.9 |
88.5±1.6 |
87.5±1.9 |
nested CV at 5% |
44.6±1.0 |
90.6±1.5 |
29.6±0.7 |
86.1±0.2 |
||
Asmare et al. [54] |
Combination of time(6), frequency(3) and perceptual features(13) |
10-fold CV |
93.9±0.4 |
94.1±1.1 |
93.7±0.8 |
93.3±0.9 |
nested CV at 5% |
66.8±2.9 |
91.5±1.9 |
52.6±3.3 |
93.5±0.7 |
||
N. K Sawant et al. [55] |
Combination of time(3), frequency(4), and perceptual features(13) |
10-fold CV |
91.7±1.3 |
90.8±1.5 |
92.6±1.2 |
92.6±1.0 |
nested CV at 5% |
62.4±2.2 |
94.7±1.6 |
46.6±2.3 |
91.7±0.7 |
||
This paper |
Combination of time(9), frequency(4), perceptual(13) and acoustic features(4) |
10-fold CV |
96.0±0.9 |
95.8±1.5 |
96.2±0.6 |
96.0±0.6 |
nested CV at 5% |
72.2±0.8 |
92.3±0.4 |
59.2±3.6 |
94.8±0.6 |
These comparative study results are discussed in the Discussion Section and the following text is inserted in lines 476-498.
During the systolic and diastolic stages of the heart cycle, RHD causes transient, non-stationary, high-frequency murmurs with low amplitude. It is critical to accurately describe abnormal cardiac murmurs to successfully detect them. As shown in Table 4, the use of purely temporal or spectral representations of the PCG as classification features was inadequate since they are limited in either temporal localization or frequency information which only achieved an f1-score of 83.2±0.8% and 86.6±1.2% for the stratified 10-fold cross-validation respectively. The modeling of heart sounds using MFCC to acquire perceptual information resulted in a reasonably good performance with an f1-score of 89.4±1.5%. Improved results were obtained by combining time, frequency, and perceptual domain features with the highest f1-score in the group being 93.9±0.4%. However, employing a unique and enhanced combination of time, frequency, perceptual and acoustic features had the best classification performance with an f1-score of 96.0±0.9%. In all the evaluations metrics using the stratified 10-fold cross-validation, the performance of this article was better.
Further comparison was done to see the performance of each feature set using the 5% prevalence rate of the study area, In this case, the biggest f1-score of 72.2±0.8% was achieved by our paper. However, the highest recall of 94.7±1.6% was recorded by N. K Sawant et al. [55] but the corresponding precision and f1-score were low. Concerning the most important metrics in a hugely unbalanced dataset, the proposed approach performed best. This is because the proposed set of features comprise not only a combination of time, frequency, perceptual features but also acoustic features which model the sensations produced by sounds using distinct elements of sound pitch and intensity.
And finally, the following text is inserted in the conclusion section. Refer to lines 559-562.
When compared to other state-of-the-art articles, our results consistently outperformed them because we were able to effectively predict murmur due mainly to RHD utilizing the proposed unique set of features.
- The authors should underline more explicitly what are the specific challenges associated with the detection of RHD compared to the more common task of abnormal heart sound detection, for which a large amount of work has already appeared in the literature. In particular, it is not clear if a machine learning method originally designed to perform abnormal heart sound detection could be adapted to the RHD detection problem, by simply training the algorithm on the proposed dataset.
Author Response: RHD is one of the most neglected diseases in the world. There are very few investigative researches that showcase the nature of valve deterioration due to RHD. It is reported that RHD may affect all the heart valves but the most frequently affected valves being the mitral valve and the aortic valve. RHD begins with mitral regurgitation, which can worsen over time as a result of persistent scarring of the valve and valve apparatus. The most frequent progression of untreated RHD with multiple episodes of ARF is from mitral regurgitation to mitral stenosis, followed by aortic regurgitation, and finally, aortic stenosis.
Some fragmented studies [1], [2],[3],[4],[5],[6], and [7] indicated that auscultation reveals the distinctive holosystolic murmur in RHD patients with mitral regurgitation, and if the mitral regurgitation is severe, an extra diastolic murmur may be heard. Rheumatic mitral stenosis develops later as the valvular scarring progresses. A low-pitched rumbling apical diastolic murmur is characteristic of mitral stenosis due to RHD. In cases where multiple valves are involved, the characteristic aortic regurgitation murmur is a high-pitched diastolic murmur. The characteristic murmur of aortic stenosis is a systolic ejection murmur, often accompanied by a diastolic decrescendo murmur if aortic regurgitation is present. These studies indicated that there is some level of uniqueness of the murmur due to RHD. For this reason, we collected data specifically from persons with RHD. As RHD murmur can span systolic and diastolic parts, we proposed our classification algorithms to considering the whole heartbeat without segmenting it to fundamental opponents. To adequately model murmur, we have also incorporated a unique set of features that span the time, frequency, perceptual, and acoustic domain features.
- Philippe Unger and Gerald Maurer. Heart valve disease: mixed valve disease, multiple valve disease, and others. The EACVI Textbook of Echocardiography, page 317, 2017.
- Liesl Zühlke, Mark E Engel, Ganesan Karthikeyan, Sumathy Rangarajan, Pam Mackie, Blanche Cupido, Katya Mauff, Shofiqul Islam, Alexia Joachim, Rezeen Daniels, et al. Characteristics, complications, and gaps in evidence-based interventions in rheumatic heart disease: the global rheumatic heart disease registry (the remedy study). European heart journal, 36(18):1115–1122, 2015.
- Maurice Enriquez-Sarano, Arsene-Joseph Basmadjian, Andrea Rossi, Kent R Bailey, James B Seward, and A Jamil Tajik. Progression of mitral regurgitation: a prospective Doppler echocardiographic study. Journal of the American College of Cardiology, 34(4):1137–1144, 1999.
- CN Manjunath, P Srinivas, KS Ravindranath, and C Dhanalakshmi. Incidence and patterns of valvular heart disease in a tertiary care high volume cardiac center: a single-center experience. Indian heart journal, 66(3):320–326, 2014.
- AS Dajani. Special writing group of the committee on rheumatic fever, endocarditis and Kawasaki disease of the council on cardiovascular disease in the young of the American heart association. guidelines for the diagnosis of rheumatic fever: Jones criteria, 1992, update. JAMA, 268:2069–2073, 1992.
- Ashraf M Anwar, Wael M Attia, Youssef FM Nosir, Osama II Soliman, Mohammed A Mosad, Munir Othman, Marcel L Geleijnse, Ali M El-Amin, and Folkert J Ten Cate. Validation of a new score for the assessment of mitral stenosis using real-time three-dimensional echocardiography. Journal of the American Society of Echocardiography, 23(1):13–22, 2010.
- Rick A Nishimura, Catherine M Otto, Robert O Bonow, Blase A Carabello, John P Erwin, Robert A Guyton, Patrick T O’Gara, Carlos E Ruiz, Nikolaos J Skubas, Paul Sorajja, et al. 2014 aha/acc guideline for the management of patients with valvular heart disease: executive summary: a report of the American college of cardiology/American heart association task force on practice guidelines. Journal of the American College of Cardiology, 63(22):2438–2488, 2014.
Minor comments:
- In lines 110-111, the authors mention the fact that cheaper and more robust strategies in the detection of RHD than echocardiography are highly sought. However, it is not clear how auscultation could represent a more robust strategy for RHD detection than echocardiography.
Author Response: It was not meant to say auscultation will be more robust. The statement is now modified.[refer to line 112-113]
Therefore, an alternative cheaper, and more robust strategy in the detection of RHD is highly sought.
- In line 138, "to have sensitivity and specificity" should be "to have low sensitivity and specificity"?
Author Response: corrected accordingly [refer to line 140]
"….to have low sensitivity and specificity"?
- In line 237 and in Table 1, there seems to be inconsistency in the number of normal subjects extracted from the PhysioNet dataset (81 vs. 111).
Author Response: Yes there were 111 unique records per subject but only 81 of them were 30 seconds or more in duration. The values are now corrected accordingly [refer to line 279-282]
- In line 246, antialiasing filtering should be applied before downsampling, in order to avoid aliasing.
Author Response: It was a statement ordering error. In the implementation, antialiasing filtering was applied before downsampling
Author Action: The preprocessing section[refer to lines 284-288] is now updated as follows.
Each record was filtered with an antialiasing bandpass filter with a frequency band from 20Hz to 1kHz. All data were then downsampled to 2 kHz. Each recording was carefully labeled as RHD and normal. Each recording was then divided into a 30 seconds duration and one record from one subject was taken. The z-score of each signal was then computed to make a zero mean and unity standard deviation
- The Lagrange multiplier alpha could be mentioned after equation (4) to avoid confusion.
Author Response: The Lagrange multiplier alpha is already mentioned just above equation 4. [refer to line 332]
- In line 295: "gaussian" -> "Gaussian".
Author Response: corrected accordingly [refer to line 339]
- In line 327: "gamma (\gamma)" -> "\gamma".
Author Response: This comment is not clear. Can you please elaborate?.
- When considering highly imbalanced training datasets, it would be interesting to consider the use of specific oversampling techniques to reduce the effect of such imbalance.
Author Response: Yes, you are right, oversampling is one way of alleviating the class imbalance. However, it also increases the likelihood of overfitting since it replicates the minority class events. In our experiments, the classifications are based on the leave-one-person-out approach; having multiple copies of one record in the process of oversampling will not be realistic. To counter the data imbalance, and to effectively model the real-life class distribution, we took repeated smaller subsets of the data taking into account the required prevalence rate.
For instance, for 5% prevalence: a single outer evaluation partition for 5% would be:1 RHD – 20 HC. The training dataset for this partition would thus be 99 RHD (100-1) and 80 HC (100-20). Oversampling the minority class of the training partition would thus actually oversample the HC’s. We believe this would be confusing as it would be the opposite of what a reader would expect when talking about low RHD prevalence rates and may also increase overfitting.
Round 2
Reviewer 1 Report
The author made careful modification and supplement. It is recommended to accept after carefully checking the English expression.
Author Response
Revision 02
Editor and reviewer comments
2021-09-24
Dear Reviewer,
We wish to thank the editor and the reviewers for your valuable time and for allowing us to submit the second revision of our manuscript, entitled ‘Rheumatic Heart Disease Screening based on Phonocardiogram’ in MDPI Sensors.
We considered the many thoughtful and constructive suggestions raised by the reviewers. We tried to address each of the comments. All modifications to the original manuscript have been marked, as part of the “manuscript with track changes” file. All modifications have been made with respect to the included single-column version of the manuscript with line numbers along the margin.
We hope that the editor and the reviewers will consider this revision for publication in MDPI Sensors, and we look forward to reading your comments and decision. We are glad to address any further questions or comments.
On behalf of all co-authors,
Sincerely,
Melkamu Hunegnaw Asmare, MSc
Department of Electrical Engineering (ESAT)
KU Leuven
Reviewer #1:
Comments and Suggestions for Authors
The author made careful modification and supplement. It is recommended to accept after carefully checking the English expression.
Author Response: Dear reviewer #3, thank you very much for the detailed review of the paper for the second time. The quality of the paper has now greatly improved.
Author Action: We used grammar checking tools to examine the writing and had the co-authors proofread it to improve the English further.
Reviewer 3 Report
In the revised version of the paper, the authors have successfully addressed all the issues raised in the previous round of reviews. Just a very minor comment stands out, that I have not explained in a sufficiently clear manner in the original review. My minor comment number 7 suggested replacing the use of "gamma (\gamma)" in line 371 with simply using the symbol "\gamma". In this sense, "\gamma" here stands for the mathematical symbol (I am using latex notation for that).
Author Response
Revision 2
Editor and reviewer comments
2021-09-24
Dear Reviewer,
We wish to thank the editor and the reviewers for your valuable time and for allowing us to submit a second revision of our manuscript, entitled ‘Rheumatic Heart Disease Screening based on Phonocardiogram’ in MDPI Sensors.
We considered the many thoughtful and constructive suggestions raised by the reviewers. We tried to address each of the comments. All modifications to the original manuscript have been marked, as part of the “manuscript with track changes” file. All modifications have been made with respect to the included single-column version of the manuscript with line numbers along the margin.
We hope that the editor and the reviewers will consider this revision for publication in MDPI Sensors, and we look forward to reading your comments and decision. We are glad to address any further questions or comments.
On behalf of all co-authors,
Sincerely,
Melkamu Hunegnaw Asmare, MSc
Department of Electrical Engineering (ESAT)
KU Leuven
Reviewer #3:
Comments and Suggestions for Authors
In the revised version of the paper, the authors have successfully addressed all the issues raised in the previous round of reviews. Just a very minor comment stands out, that I have not explained in a sufficiently clear manner in the original review. My minor comment number 7 suggested replacing the use of "gamma (\gamma)" in line 371 with simply using the symbol "\gamma". In this sense, "\gamma" here stands for the mathematical symbol (I am using latex notation for that).
Author Response: Dear reviewer #3, thank you very much for the detailed review of the paper for the second time. The quality of the paper has now greatly improved.
Author Action. The term gamma is removed and replaced with γ. We used grammar checking tools to examine the writing and had the co-authors proofread it to improve the English further.